# Parasitic structure defect blights sustainability of cobalt-free single crystalline cathodes

Lei Yu[1,7], Alvin Dai[2,7], Tao Zhou[1], Weiyuan Huang[2], Jing Wang[2], Tianyi Li[3], Xinyou He[4], Lu Ma[5], Xianghui Xiao[5], Mingyuan Ge[5], Rachid Amine[6], Steven N. Ehrlich[5], Xing Ou[4]✉, Jianguo Wen[1]✉, Tongchao Liu[2]✉ & Khalil Amine[2]✉

Recent efforts to reduce battery costs and enhance sustainability have focused on eliminating Cobalt (Co) from cathode materials. While Co-free designs have shown notable success in polycrystalline cathodes, their impact on single crystalline (SC) cathodes remains less understood due to the significantly extended lithium diffusion pathways and the higher-temperature synthesis involved. Here, we reveal that removing Co from SC cathodes is structurally and electrochemically unfavorable, exhibiting unusual voltage fade behavior. Using multiscale diffraction and imaging techniques, we identify lithium-rich nanodomains (LRNDs) as a heterogeneous phase within the layered structure of Co-free SC cathodes. These LRNDs act as critical tipping points, inducing significant chemo-mechanical lattice strain and irreversible structural degradation, which exacerbates the voltage and capacity loss in electrochemical performance. Our findings highlight the considerable challenges of developing Co-free SC cathodes compared to polycrystalline ones and emphasize the need for new strategies to balance the interplay between cost, sustainability, and performance.

The rapid adoption of electric vehicles (EVs) offers a promising way to reduce carbon emissions into the environment and contribute to the long-term goal of carbon neutrality[1–3]. Nevertheless, the excessive utilization of cobalt (Co) in battery raw materials raises concerns about the sustainability of EVs over time, owing to the scarcity of Co, its elevated costs, and volatile supply chains[4–7]. In response, developing Co-free cathode materials has gained momentum, aiming to reduce Co dependence, ensure robust capacity, and enhance battery sustainability[8–12]. Nowadays, polycrystalline Co-free cathodes have demonstrated competitive electrochemical performance compared to conventional Co-containing cathode materials, which raises the question of whether Co use is even necessary[13–17]. However, regardless of whether Co is used, there have been inherent limitations that persist within the long-standing design of polycrystalline spherical particle morphology. These micron-sized secondary particles, comprising multiple primary nanoparticles, permit electrolyte penetration into the bulk through interstitial gaps among crystal grains to initiate severe interfacial side reactions[18–20]. Moreover, the random crystallographic orientation of primary particles will exacerbate mechanical degradation due to anisotropic volume evolution during electrochemical cycling[21–23].

Single crystalline (SC) LiNi$_x$Mn$_y$Co$_z$O$_2$ (NMC) holds the potential to extend the lifespan of lithium-ion batteries (LIBs) by mitigating

[1]Center for Nanoscale Materials, Argonne National Laboratory, Lemont, IL, USA. [2]Chemical Sciences and Engineering Division, Argonne National Laboratory, Lemont, IL, USA. [3]X-ray Science Division, Advanced Photon Sources, Argonne National Laboratory, Lemont, IL, USA. [4]Engineering Research Center of the Ministry of Education for Advanced Battery Materials, School of Metallurgy and Environment, Central South University, Changsha, P. R. China. [5]National Synchrotron Light Source II, Brookhaven National Laboratory, Upton, NY, USA. [6]Material Science Division, Argonne National Laboratory, Lemont, IL, USA. [7]These authors contributed equally: Lei Yu, Alvin Dai. ✉e-mail: ouxing@csu.edu.cn; jwen@anl.gov; liut@anl.gov; amine@anl.gov

challenges such as surface side reactions and mechanical degradation found in polycrystalline NMC[24–26]. Similarly, there is motivation to reduce Co content in SC-NMC and realize Co-free SC cathodes to reduce cost challenges, as well as resolve sustainability issues related to material sourcing. While the success of Co-free cathode in polycrystalline is indeed inspiring, the thorough understanding for the impact of Co-free designs on SC-NMC has yet to be detailed. Fundamentally, the removal of Co and its impacts on dynamic structure evaluation in both the synthesis and electrochemical operation are unclear. Co plays a pivotal role in facilitating Li$^+$ diffusion within the lattice, a particularly critical attribute for SC-NMC cathodes, given their elongated diffusion pathways whereby heterogeneous Li$^+$ diffusion can occur and lead to strain evolution[27,28]. In addition, the fabrication of SC morphology using high-temperature conditions intrinsically introduces substantial structure defects. Addressing such defects has historically relied on Co, which, by virtue of its tendency to oxidize preferentially and unique electronic configuration, promotes structural ordering and suppresses the formation of Ni/Li mixing[22,29–31]. Therefore, notwithstanding the inherent cost benefits, the merits of Co-free designs in SC-NMC remain a mystery, necessitating a more profound and comprehensive understanding.

In this work, we systematically investigate structures of Co-free high-Ni SC cathode materials to understand the impact that removing Co will have. It was found that removing Co is electrochemically and structurally unfavorable for SC cathodes and significantly affects electrochemical properties and structural integrity. Leveraging multiscale structural characterizations, we revealed that the removal of Co and the high-temperature calcination to form SC morphology would introduce unexpected atomic Li-rich domains that are randomly distributed in the bulk lattice, which was rarely observed in both Co-containing NMC cathodes and polycrystalline Co-free cathodes. In addition, given the extended diffusion pathways of SC, Li diffusion gradients are exacerbated by the presence of Li-rich domains, which can lead to significant chemo-mechanical lattice strain that was directly confirmed by synchrotron-based X-ray nanodiffraction and TEM characterization. As a result, the Co-free SC cathode displays unsatisfactory initial capacity, coulometric efficiency, and unusual voltage degradation. This work underscores that the realization of Co-free SC cathodes is a formidable challenge that requires substantial advancements before it can materialize, particularly in the absence of a suitable alternative element.

## Results

### Initial structure and electrochemical performance

A series of Co-free Ni-rich cathodes with different Ni contents were synthesized using a two-step method involving co-precipitation synthesis of the precursors and solid-state calcination. The detailed preparation procedure can be found in the experimental method section. Here, LiNi$_{0.75}$Mn$_{0.25}$O$_2$ (SC75) was selected as a model sample for the systematic investigations because this optimal composition is supposed to be more stable than ultrahigh Ni content (>80%) cathode and provide higher energy density than median Ni-content (50%–60%) cathode while being more cost-efficient. The initial morphology of the SC75 sample after synthesis was first investigated by scanning electron microscope (SEM). As shown in Fig. 1a, the SC75 particles present a fairly uniform particle size in the range of 3–5 µm. The particle size is apparently larger than that of the primary particle of conventional polycrystalline cathode materials, which may cause sluggish Li insertion and the formation of initial structure defects during the sample synthesis[28]. The scanning transmission electron microscopy coupled with energy-dispersive X-ray spectroscopy (EDS) elemental analysis further shows that the sample was well synthesized with an ideal compositional ratio (Supplementary Fig. 1). The phase structure of SC75 was then investigated by synchrotron-based high-energy X-ray diffraction (HEXRD) and Rietveld refinement analysis. As illustrated in Fig. 1b, all Bragg diffraction peaks

can be well indexed to a hexagonal α-NaFeO$_2$-type layered structure (R-3m space group) without any impurity detected. It should be noted, however, that the (003) peak intensity and the ratio of (003)/(104) peak are relatively low, which together suggests the SC75 has a relatively high content of Li/Ni disorder or structure defects. This is further confirmed by Rietveld refinement analysis as shown in Supplementary Table S1, where the Li/Ni disorder is quantified as 5.44%. This is higher than most SC-NMC cathode materials but similar to Co-free compositions[24,32]. It has been well documented that Co is favorable for formation of well-layered structures, so it is expected that removing Co would result in more Li/Ni disordering[31].

The electrochemical performance of SC75 was evaluated by coin-type cells with as-prepared SC75 cathode (The mass loading is ~5.2 mg cm$^{-2}$.) and Li metal anode at different voltages. The galvanostatic charge/discharge curves in Fig. 1c show that the initial capacity was recorded as 185 mAh g$^{-1}$ at 0.1 C (20 mA g$^{-1}$) within a voltage window of 2.8–4.4 V. It is of note that the coulombic efficiency of the initial cycle was only 84%, which is lower than those of typical Co-containing SC-NMC cathodes[33–35]. The subsequent cycles operated at 0.5 C (100 mA g$^{-1}$) show a slight capacity degradation. The corresponding dQ/dV curves are used to describe the structural reversibility during electrochemical cycling. As shown in Fig. 1d and Supplementary Fig. 2a, the presented dQ/dV peaks reflect the phase transitions of the layered structure at different stages with a sequence of H1-M-H2-H3 transitions in charging and inverse transitions in discharging[36]. During cycling, the peaks at 4.2 V clearly shift to lower voltage with degraded peak intensity, while the peak at 3.75 V is relatively stable. This suggests that capacity fade and structure degradation mainly occur during the H2–H3 phase transition at high voltage.

Figure 1e further shows the electrochemical performance of the SC75 electrode when the upper cut-off voltage is raised to 4.6 V. The initial capacity is recorded as 194 mAh g$^{-1}$, a bit higher than that of 4.4 V. However, the coulombic efficiency and rate performance are even worse compared to that at 4.4 V, suggesting an additional reaction occurred at high voltage. This is further confirmed by the charge/discharge profiles and the corresponding dQ/dV curves (Fig. 1f and Supplementary Fig. 2b). Besides the fast capacity fading, an obvious voltage decay can be observed in the prolonged cycles, indicating an increase in impedance. In Fig. 1f, the dQ/dV curves show significant peak shifts during charge and discharge as long-term cycling proceeds. For example, the peaks at both 4.2 V and 3.7 V have quickly shifted to lower potential with a weakened intensity. In addition, new peaks at 3.90, 3.77, and 3.71 V are present upon discharge after 100 cycles, indicating an irreversible phase transition occurred when cycled at 4.6 V. In general, the Co-free SC75 electrode exhibits underachieved cycle stability and fast structure degradation at high operation voltage.

The capacity and voltage retentions are summarized in Fig. 1g, h, respectively. The capacity retention after 100 cycles at 4.4 V is 85% of the initial capacity, whereas it is only 75% at 4.6 V. The exacerbated capacity decay at 4.6 V should be attributed to the irreversible structural transformations observed in the dQ/dV peak profiles. Moreover, as displayed in Fig. 1h, the SC75 electrode cycling at 4.6 V shows an accelerated voltage fade, where the voltage fade is recorded as −0.9 mV per cycle at 4.4 V but increases to −2.2 mV per cycle at 4.6 V. Considering unexpected low coulombic efficiencies, poor structural reversibility, and voltage fade, it is highly suspected that the removal of Co may have led to subtle structural changes in SC75 that impact electrochemical performance and structure integrity.

### Structural and chemical stability by macroscale characterizations

To uncover underlying structural factors, in-situ HEXRD was conducted to monitor the structure reversibility during the electrochemical process (Supplementary Figs. 3, 4). Figure 2a, b depict the two-dimensional

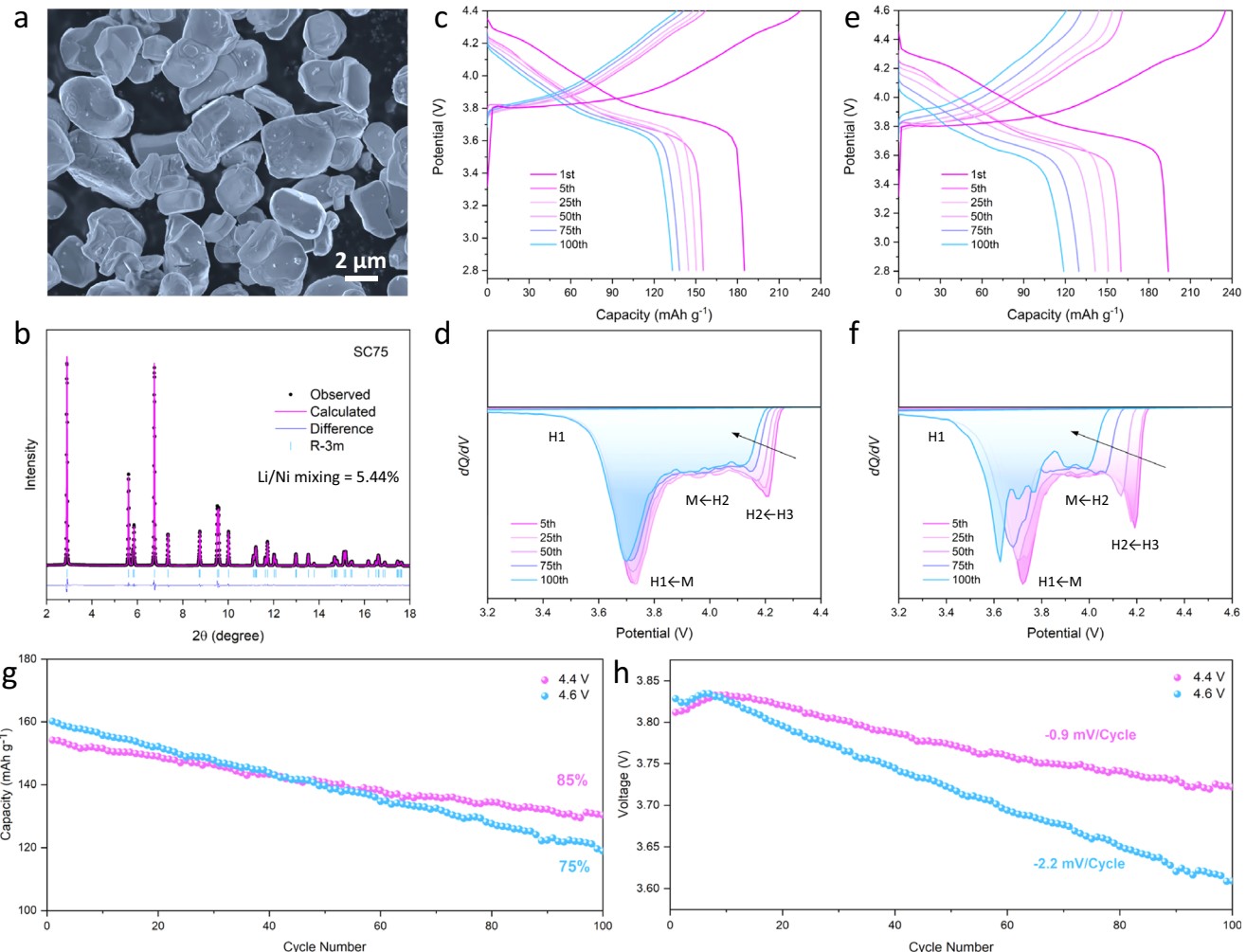

**Fig. 1 | Structure and electrochemical performance of SC75 cathode. a** SEM image of the prepared SC75 particles. **b** HEXRD and Rietveld refinement plot of the SC75 sample. **c** Charge and discharge curves of SC75 electrode within a cutoff voltage range of 2.8−4.4 V. **d** Associated *dQ/dV* profiles for 4.4 V cycling. **e** Charge and discharge curves of SC75 electrode within a cutoff voltage range of 2.8−4.6 V. **f** Associated *dQ/dV* profiles for 4.6 V cycling. **g** Cyclic stability curves of SC75 electrodes at the current rate of 0.5 C (100 mA g⁻¹) within the voltage ranges of 2.8−4.4 V and 2.8−4.6 V after three activation cycles. **h** Detailed voltage fade analysis during the long-term cycling. The mass loading of active material in electrode is ~5.2 mg cm⁻².

(2D) contour plots for the structural evolutions of the SC75 electrodes during the first charge-discharge cycle at the different voltage ranges of 2.8−4.4 V and 2.8−4.6 V, respectively. In general, the in-situ HEXRD results operated at 4.4 V and 4.6 V show similar phase transition behavior without additional phase transition detected at high operating voltage. Specifically, the (003) reflections initially shift to lower *2θ* values, which indicate increases in *c-axis* lattice spacing and correspond to H1-M and M-H2 transitions. After that, the (003) reflections undergo a sudden right shift, implying rapid decreases in *c-axis* lattice spacing, which is associated with the H2−H3 phase transitions at high voltages. During the discharge process, all the peaks smoothly shift back to the original position following a reverse sequence. Compared with 4.4 V, the SC75 cathode cycling at 4.6 V also exhibits a reversible structure evaluation but slightly larger change in the *c-axis* lattice parameter, which is attributed to more Li removal with increased cutoff voltage. Generally, both in-situ HEXRD results seem to suggest that the structure evolution of SC75 is reversible without obvious structure degradation during high-voltage operation.

X-ray absorption spectroscopy (XAS) was further conducted to investigate the chemical state changes for Ni and Mn at different charged and discharged states. As shown in Fig. 2c, it is clear that the peak position of the Ni K-edge shows a positive shift toward high

absorption energy upon charging, reflecting an increase in the oxidation state. When discharged to 2.8 V, the profile of the Ni K-edge returns to the initial state, indicating a good chemical reversibility during electrochemical operation. In Fig. 2d, the X-ray absorption near edge structure (XANES) of the Mn K-edge does not show apparent shifts in the electrochemical process, which illustrates that the Mn valence state remains stable and electrochemical inactive. The shape deformation of the curve at 4.4 V can be related to the change in local coordination environment. Overall, the above macroscale characterizations suggest that the structural and chemical evolution of SC75 seem reversible. However, the unusual capacity/voltage degradation and apparent structural instability observed in the electrochemical tests remain unexplained. Clearly, macroscopic characterizations may not be sufficient to identify the structural changes associated with the removal of Co. Therefore, microscopic structural characterizations at the nano or atomic scales are essential to understanding the SC75 sample.

## 3D continuous rotation electron diffraction and microscopic imaging observation

To thoroughly investigate the microstructure of SC75, 3D continuous rotation electron diffraction (3D-CRED) was utilized to analyze the

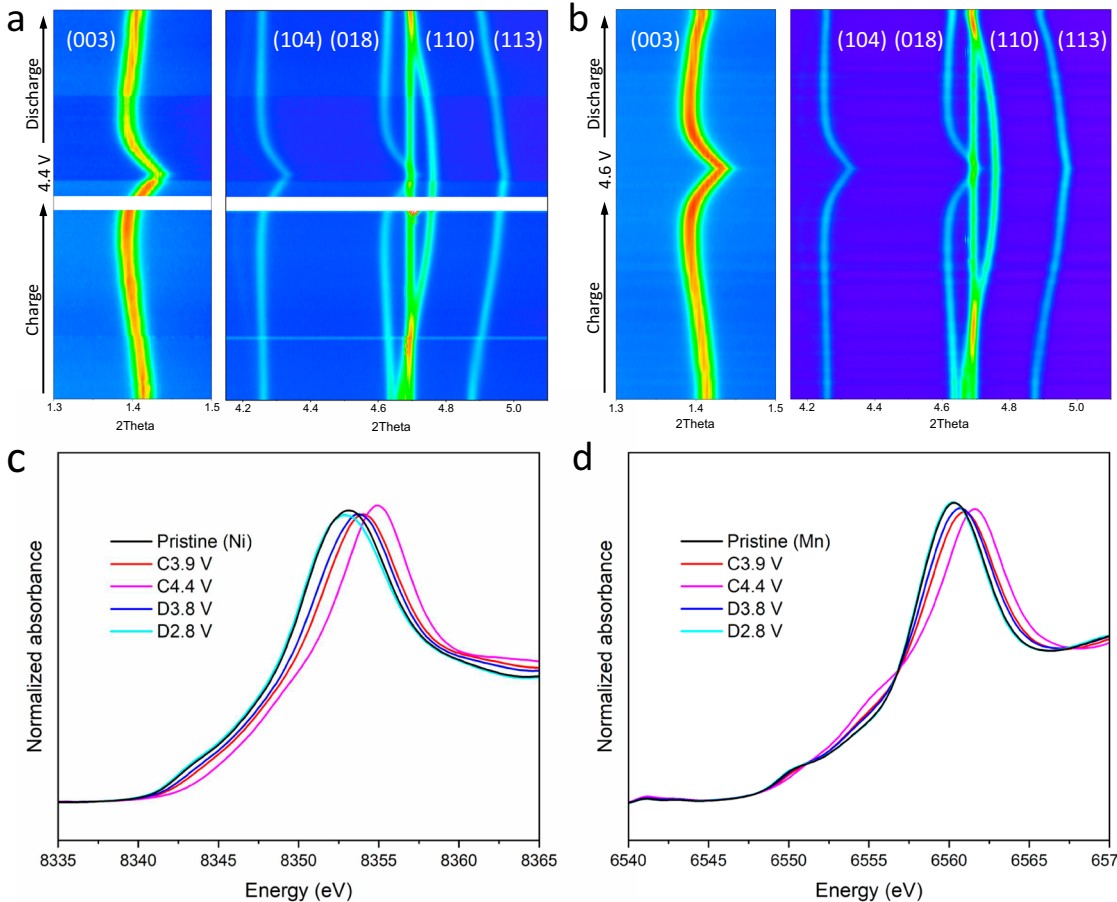

**Fig. 2 | Statistical structure and valence evolution of SC75 during charge-discharge process. a, b** Two-dimensional contour plots of in situ HEXRD for the structural evolution of SC75 at the initial cycle within the voltage ranges of 2.8–4.4 V (**a**) and 2.8–4.6 V (**b**). **c, d** Ex situ Ni (**c**) and Mn (**d**) K-edge XANES spectra of SC75 at different charge/discharge states.

lattice structure and distortion at particle level. Unlike conventional selected area electron diffraction (SAED) and polycrystalline XRD, where only the two-dimensional diffraction information can be obtained, 3D-CRED collects a tilt series of diffraction patterns by continuously single-tilting the sample to reconstruct the 3D reciprocal lattice (Fig. 3a)[18,37,38]. Hence, the structural information can be resolved in arbitrary directions virtually instead of a specific direction in the conventional transmission electron microscope (TEM) observation. Furthermore, compared to single-crystal XRD, the 3D-CRED technique allows the data to be acquired on much smaller sample volumes and collected within minutes. Therefore, this method is especially suitable for studying the structure distortion of micron-sized SC cathode particles.

Randomly selected SC75 single particles were used for the 3D-CRED measurement. As shown in Supplementary Figs. 5, 6, the samples were adjusted at the eucentric height under the illumination of a parallel electron beam during the collection of the 3D-CRED dataset. Figure 3b, c and Supplementary Fig. 5b are the projection views of the reconstructed 3D reciprocal lattice after data processing, which is different from the conventional 2D-SAED pattern that is just an intersection of the Ewald sphere with the 3D reciprocal lattice. Supplementary Fig. 5b displays the reconstructed 3D diffraction pattern obtained from the 3D reciprocal lattice projection along the $c^*$ direction. The regular arrangement of diffraction spots implies an ordered intralayer structure. However, the cyan-color diffraction spots, which are not expected in a conventional layered LiTMO$_2$ structure (TM, transition metal), have been observed, indicating the presence of a superlattice structure. The reconstructed 3D-reciprocal lattice

projections along the $a$ and $a^*$ directions reflect the interlayer structure information (Fig. 3b, c). Multiple cyan-color streaking lines along the $c^*$ direction are observed, and this kind of diffraction feature corresponds to the stacking faults of layered structure in the real space[39,40]. As displayed in Supplementary Fig. 6, the other detected SC75 particles also present similar 3D-CRED results, ruling out the occasionality of the results. The 3D-CRED technique reveals new structural defects in SC75 that are invisible in common NMC cathode (Supplementary Fig. 7) and are also undetectable with the bulk characterization techniques. This indicates that the microstructure of SC75 is more complex than conventional NMC materials, providing some clues for the undesirable electrochemical behavior.

Following the guidance from 3D-CRED, detailed TEM characterization of the SC75 particle was carried out at different zone axes to investigate the exact microstructure at the nanoscale. Supplementary Fig. 8 shows low-magnification TEM images at the zone axes of [110], [210], and [16 8 $\bar{1}$] taken at the same location. The [110] zone axis is a basic orientation of the layered LiTMO$_2$ structure. The corresponding SAED pattern shown in Fig. 3d manifests a standard diffraction feature of layered LiTMO$_2$ structure without impurity phase. Moreover, the HRTEM image in Fig. 3g further displays the atomic arrangement of the layered structure, where the TM layers can be clearly seen with no obvious intensity difference along this orientation. The slightly enhanced contrast in Li layers implies a small amount of Li/Ni mixing, consistent with the XRD result. The TEM observation at this zone axis does not show the stacking faults and superlattices that are observed in 3D-CRED. This also reflects that these structural defects are easily overlooked in conventional characterizations.

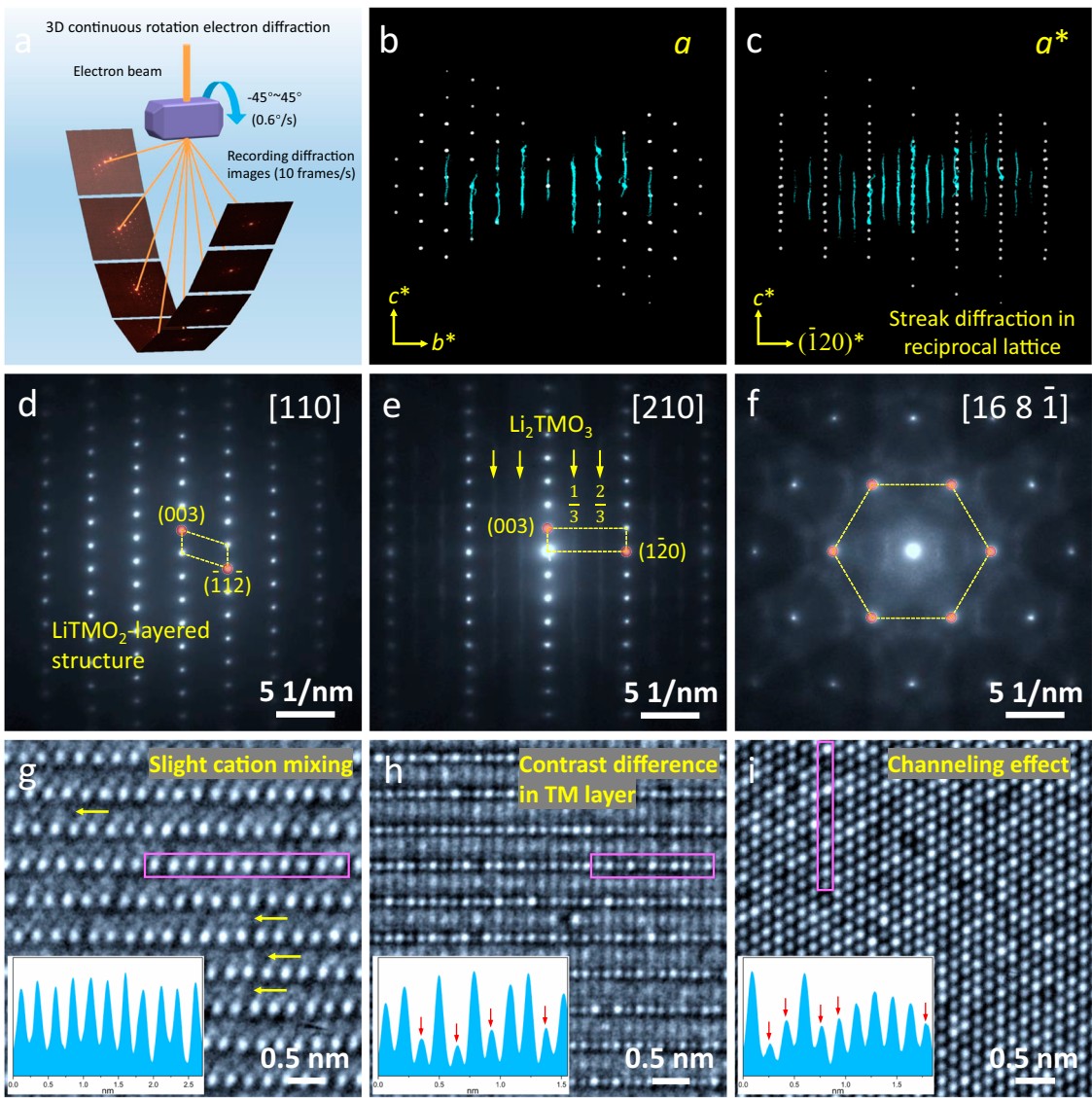

**Fig. 3 | 3D-CRED, 2D-SAED, and HRTEM characterization. a** Schematic diagram of the 3D-CRED technology. **b, c** The reconstructed 3D-reciprocal lattice projections along the $a$ (**b**) and $a^*$ (**c**) directions. **d–f** 2D-SAED patterns and **g–i** corresponding HRTEM images along the [110] (**d, g**), [210] (**e, h**) and [16 8 $\bar{1}$] (**f, i**) zone axes, the insets in HRTEM images are the corresponding line intensity profiles of magenta boxes.

When tilting to the [210] zone axis, the TM atoms are arranged vertically along the $c$ axis between the layers and have smaller in-plane spacing. Hence, higher-resolution imaging is required to analyze structure details. As shown in Fig. 3e, the corresponding SAED pattern at the [210] zone axis presents additional diffraction steaks parallel to (003) direction at n/3 ($\bar{1}$20) positions, which is a typical diffraction feature of the $Li_2TMO_3$ (Li-rich phase, Supplementary Fig. 9), corresponding to the ordering of excess Li in the TM planes. The HRTEM image in Fig. 3h further demonstrates the SAED result, and the apparent contrast differences are observed in the TM planes owing to the occupation of Li (Supplementary Fig. 10). These results indicate the layered structure of SC75 is mixed with the small amount of $Li_2TMO_3$ phase, which is rarely observed from typical NMC cathode materials (Supplementary Fig. 11). The same structural features are also present in SC80 and SC90 samples, suggesting that this is a common phenomenon for Co-free SC cathodes (Supplementary Figs. 12,13).

The above TEM characterization at the interlayer [210] zone axis reveals the presence of Li/TM ordering in the TM plane. Hence, the intralayer observation at the [001] zone axis is desirable to reflect the distribution of the impurity phase. However, the angle of 90° between

these two directions makes TEM observation inaccessible at [001] zone axis on the same particle, owing to the limited tilt angles of a double tilt TEM holder (typically ±40°). As shown in Fig. 3f, the same TEM observation was chosen to be performed at [16 8 $\bar{1}$] zone axis, which is reached by tilting the sample at [210] zone axis around the ($\bar{1}$20) by 20°. This zone axis is equivalent to the [001] zone axis in the layered structure because they both belong to the family of <111> crystal directions in the parent structure of the face-centered cubic lattice. The SAED pattern in Fig. 3f displays the typical sixfold diffraction spots but with a weak superlattice. In Fig. 3i, the HRTEM image exhibits the apparent channeling contrast difference between columns. In channeling theory, high-energy electrons travel along atomic columns. Each atom in the column behaves as a mini-lens to bend high-energy electrons, such that light elements bend electrons less compared to heavy elements. Therefore, the intensity variation in the same type of atomic columns reflects the occupation rate change. This observation in Fig. 3i indicates that the $Li_2TMO_3$ phase is randomly distributed and remains extremely small at atomic scale[41]. Hence, the heterogeneous $Li_2TMO_3$ phase can be seen as numerous Li-rich nanodomains (LRNDs) embedded in the layered structure. This result agrees with the prior

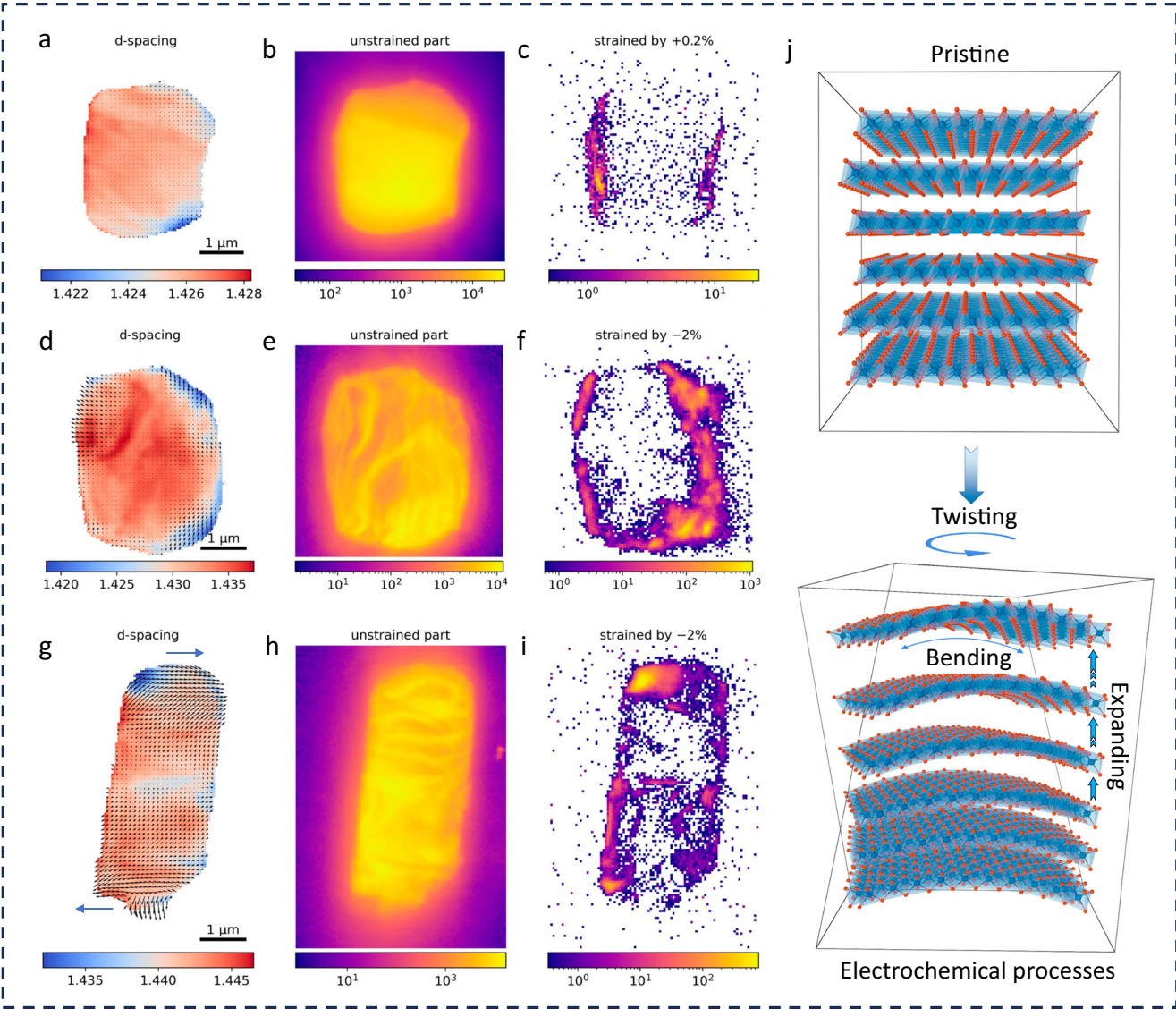

**Fig. 4 | Scanning X-ray nanodiffraction microscopy characterization.**
**a, d, g** Structural mapping of SC75 particles at pristine (**a**), charged (**d**), and after 100 cycles (**g**) stages based on 5D-diffraction datasets. **b, e, h** The corresponding integration of unstrained parts. **c, f, i** The corresponding integration of strained parts. **j** the comparison of structural models for SC75 before and after cycling.

bulk XRD observations where no superlattice peaks were detected due to the extremely small domain size. Noted that the $Li_2TMO_3$ is electrochemically and kinetically unfavorable, so that its domain will directly affect the electrochemical properties of SC75.

**LRNDs cause lattice strain and lattice tilt**
The unstable $Li_2TMO_3$ has been considered as the Achilles' heel of Li and Mn-rich cathodes, but its impacts on SC cathodes remains unknown, as it is firstly observed in Ni-based layered oxide cathodes. Given only limited content of atomic-scale LRNDs observed, it likely affects microstructure evolution rather than occurring at macroscale level. To delve into this, scanning X-ray diffraction microscopy (SXDM) with high spatial resolution was performed to analyze the lattice strain evolution of SC75 particles. To increase the strain sensitivity, dark field images are created by integrating all the pixels that fall in a certain $q$-range in the reciprocal space based on the obtained five-dimensional diffraction dataset (see Methods)[42]. Figure 4a, d, g shows the structural mappings of SC75 particles at different electrochemical states, where the color gradient reflects the lattice parameter variation (lattice strain) and the arrow indicates the deformation of the lattice that

manifests as lattice tilt. Note that lattice tilt here refers to rigid body rotations of the crystal lattice which is different from the effect of strain (Fig. 4j). The direction of the arrows shows the direction of the tilt, and the length of the arrows is indicative of the magnitude of the tilt. Figure 4a shows that the pristine particle has a narrow $d$-spacing distribution with only 0.25% variation and negligible lattice bending (maximum amplitude of 5 mrad). In the dark field image for the unstrained part (i.e., strain of ~0%, Fig. 4b), the largely uniform intensity contrast indicates that the pristine particle is a high-quality single crystal with no visible defects. A small portion of the surface is strained by only 0.2% (Fig. 4c), which may be caused by the slightly off-stoichiometric Li content on the particle surface.

When the particle was charged, the lattice parameter variation and lattice bending exhibit significant changes. As shown in Fig. 4d, both the lattice variation (lattice strain) and lattice bending at this stage are double compared to the pristine particle. More considerable bending was again found in the areas with larger variations of $d$-spacing, and the maximum magnitude of bending has more than doubled to 13 mrad. The dark field image on the unstrained area (Fig. 4e) shows an inhomogeneous distribution. More importantly, a

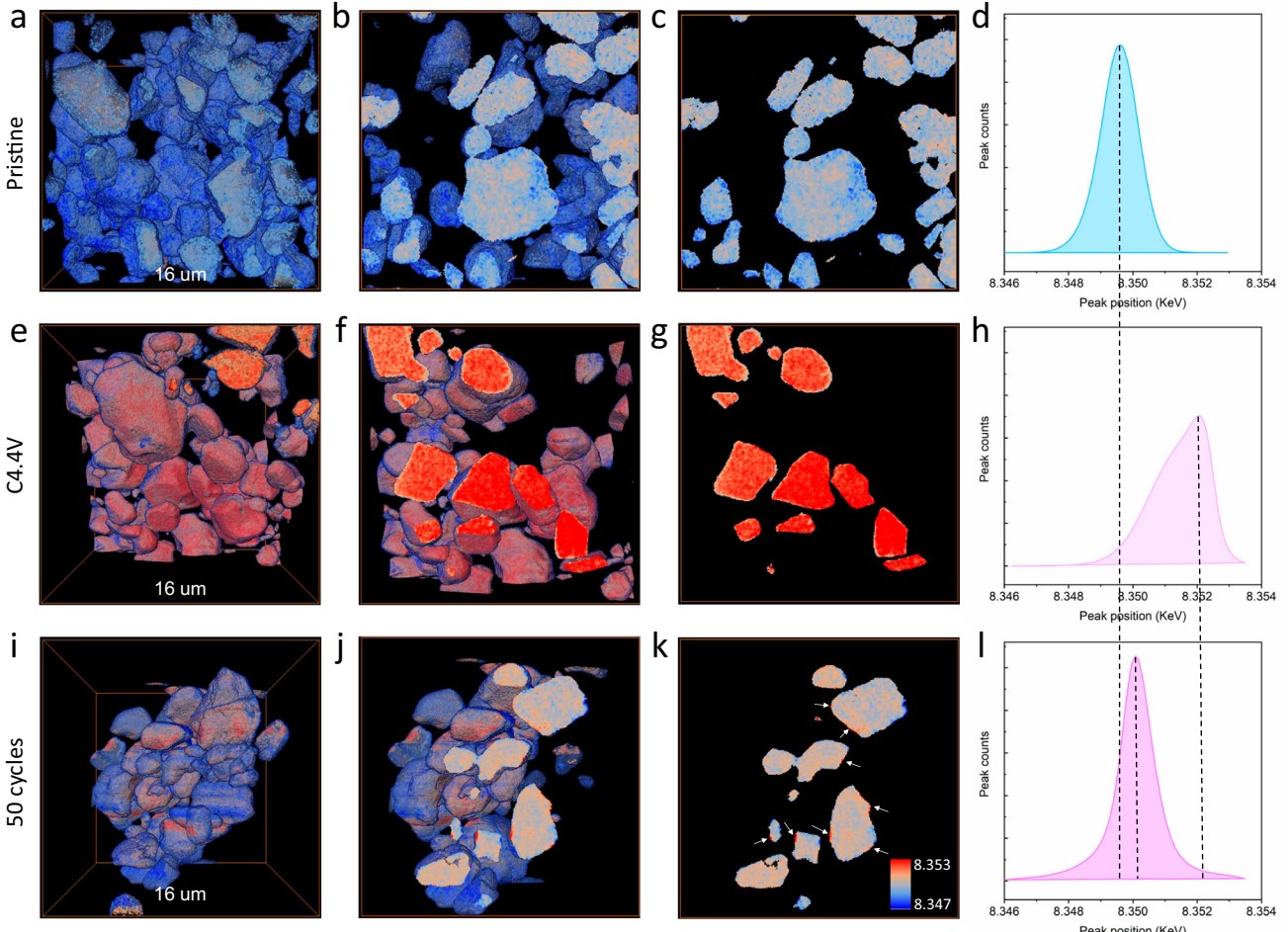

**Fig. 5 | 3D TXM-XANES for chemical state distribution. a–d** The 3D TXM-XANES mapping (**a**), cross-sectional views (**b**, **c**), and statistic distribution (**d**) based on whiteline peak position for pristine SC75. **e–h** The 3D TXM-XANES mapping (**e**), cross-sectional views (**f**, **g**), and statistic distribution (**h**) based on whiteline peak position for charged SC75. **i–l**, The 3D TXM-XANES mapping (**i**), cross-sectional views (**j**, **k**), and statistic distribution (**l**) based on whiteline peak position for cycled SC75.

significant amount of surface area was shown (Fig. 4f) to have a large compressive strain of 2%, which is ten times larger than that of the pristine particle. The surface compressive strain is highly related to the inhomogeneous delithiation caused by the faster reaction rate of the surface and sluggish Li$^+$ diffusion in the bulk. Also, the obvious $d$-spacing differences also exist within the bulk. Both heterogeneous electrochemical processes should be attributed to the sluggish kinetics of Li$_2$TMO$_3$ domains and its differential electrochemical activities with LiTMO$_2$. This strain induced by the heterogeneous electrochemical process will continuously accumulate during the repeated cycles.

This is further confirmed by the lattice strain and lattice bending analysis conducted for the cycled sample. As shown in Fig. 4g, compared to the pristine state, the much larger average lattice parameter of the cycled SC75 suggests that a certain amount of Li could not reinsert into the lattice. This indicates that the SC75 undergoes severe structural irreversibility during long-term cycling, which is further corroborated by the $d$-spacing variation where the inhomogeneous $d$-spacing has been expanded into the particle bulk. In addition, large lattice bending was observed on most parts of the particle, with a maximum amplitude of 30 mrad, which is six times higher than the pristine state. Moreover, the top part of the particle is tilted increasingly to the right, while the bottom part is tilted increasingly to the left, indicating a twisting of the entire particle

around its long axis. An inhomogeneous distribution is again observed in the unstrained dark field image (Fig. 4h), with some parts compressively strained to 2% (Fig. 4i). Based on the above understanding, a vivid model was built to reflect the structural change of the SC75 after cycling. As shown in Fig. 4j, two types of mechanical change behaviors including lattice strain (expansion/contraction) and lattice tilt (bending, and twisting) can be distinguished, and their combination induces strong structural changes and degradations.

The above SXDM experiment reveals the heterogeneous lattice structure changes within the particles. Furthermore, full-field transmission X-ray microscopy (TXM) coupled with 3D XANES was performed to analyze the variation of chemical states at the particle level. The uniform color distribution on both 3D TXM-XANES mapping and cross-sectional 2D TXM-XANES mappings (Fig. 5a–c) demonstrates a well-distributed Ni-related phase and the homogeneous oxidation state of the Ni element in the pristine SC75. The statistical analysis of the whiteline peak position shows a symmetric distribution and is centered at 8349.6 eV (Fig. 5d). During charging, the observed color in the particles turns into red, indicating the increase of valence state of Ni with Li removal (Fig. 5e–g). However, the uneven color distributions imply the heterogeneous electrochemical process, which can be attributed to sluggish Li$^+$ diffusion caused by presence of LRNDs. These phenomena are further reflected in the statistical whiteline peak position curve, which shows a positive shift but highly asymmetric

distribution (Fig. 5h). The inhomogeneous distribution of chemical states persists in the long-term cycled sample even at discharge states (Fig. 5i–k). Especially, the near-surface region displays the elevated valence of Ni (Fig. 5k, l), suggesting that some of the Li is not reinserted into the lattice due to the lattice distortion and phase transition, in agreement with the SXDM observation.

The morphology and structure changes after cycling were investigated by TEM. Figure 6a shows the low-magnification TEM image of the cycled SC75 particle at 4.4 V, where no obvious morphology damage is observed. Figure 6b is the corresponding SAED pattern. It is clear that the sample, after 100 electrochemical cycles, still maintains the basic layered structure. However, the $Li_2TMO_3$ phase is significantly reduced with the generation of a new spinel phase (Supplementary Fig. 9). Furthermore, the elongated diffraction spots, especially at high order, indicate the presence of lattice strain inside the particles after cycling at 4.4 V. The microscopic structural changes after cycling were further revealed by the HRTEM measurements. It is observed that multiple moire patterns occur in the layered structure, indicating the localized lattice variations caused by strain concentration (Fig. 6c and Supplementary Fig. 14). In Fig. 6d, the atomic-resolution HRTEM image again demonstrates the microstructure transformation of the spinel phase in the bulk area. In principle, the $Li_2TMO_3$ phase is electrochemically inactive below 4.5 V. However, the localized strained area facilitates oxygen loss from the lattice, thereby triggering the TM migration and phase transition. Hence, these results clearly indicate that the LRNDs in SC75 induce structural degradation as well as the accumulation of lattice strain during the electrochemical process at 4.4 V, consistent with the SXDM results. Furthermore, it is worth mentioning that the bulk structural degradation caused by LRNDs is different from the surface reconstruction mechanism resulting from interfacial side reactions with electrolyte.

Figure 6e further exhibits the low-magnification TEM image of the SC75 particle after cycling at 4.6 V. Unlike cycling at 4.4 V, severe intragranular cracks occur, leading to particle damage (Supplementary Fig. 15), which is the result of internal strain release by mechanical degradation. The SAED pattern in Fig. 6f indicates the complete disappearance of the $Li_2TMO_3$ phase and the substantial increase of the rock salt phase after long-term cycles (Supplementary Fig. 9). As shown in Fig. 6g, the HRTEM image characterizes the microstructure near the crack inside the particle. It is clear that the structure at the crack has transformed into a rock-salt phase, indicating the Li layer is completely occupied by the TM atom. The rock salt phase is electrochemically inactive, thus leading to a significant capacity decrease as observed in the electrochemical tests. Overall, the root cause of capacity fading is attributed to crack initiation by strain evolution, which exposes fresh surfaces with electrolytes to trigger irreversible phase transitions of rock salt. According to our previous report, the strain concentration during the electrochemical process is closely related to the voltage fade and anion charge compensation mechanism of the $Li_2TMO_3$ phase[43]. As shown in the line-scan EELS of Fig. 6h, this is also well evidenced by the reduction of the prepeak of O K-edge on the bulk phase transition area[44]. Hence, the TEM results for the cycled SC75 sample indicate that the randomly distributed LRNDs are like structural faults buried in the $LiTMO_2$ phases, which are completely excited under high voltage to cause structure destruction.

### Formation and degradation mechanism of the LRNDs

The formation of the LRNDs, where Li is present in the TM layer, can be attributed to the magnetic interactions within TM layers and the rigorous calcination conditions necessary for SC cathode synthesis. The magnetic interaction mechanism has recently been recognized as a critical driving force in tuning the atomic occupancies of TM and Li ions in layered cathode materials[45–49]. In a typical NMC cathode (Fig. 6i), the spins of TM ions construct a two-dimensional triangular network.

Within this network, magnetic $Ni^{2+}$ and $Mn^{4+}$ ions will induce strongly frustrated magnetic interactions in the TM layer, giving rise to lattice instability. This can be mitigated by introducing nonmagnetic $Co^{3+}$ ions to form a nonmagnetic center on a honeycomb lattice unit, stabilizing the surrounding TM spins in an antiferromagnetic arrangement[31]. Similarly, Li ions which also has no spin, can occupy similar positions to $Co^{3+}$ in TM layers to alleviate magnetic frustration and form a stable cathode structure. As a result, in the case of Co-free SC75 studied here, some Li ions preferentially occupy positions in the TM layers forming $Li_2MnO_3$ phase to release the strong magnetic frustration due to Co's absence. On the other hand, the increased calcination temperature and grain size greatly exacerbate the uneven distribution of Li in the SC particles. The combination of these two aspects finally leads to the formation of the LRNDs (Fig. 6i).

As shown in Fig. 6j, the presence of LRNDs in Co-free SC cathodes shows detrimental effects on the electrochemical performance. Specifically, LRNDs exhibits differential electrochemical reactivities from $LiTMO_2$ phases and sluggish $Li^+$ diffusion. This discrepancy results in heterogeneous electrochemical behavior and varied lattice structure evolution within the coherent lattice. The combination of these effects leads to severe lattice strain in localized areas, causing gradual oxygen release and irreversible phase transitions to spinel in the particle bulk, ultimately resulting in substantial capacity loss. As a result, the electrochemical performance of Co-free SC cathodes shows rapid capacity loss, performing notably worse than cobalt-containing cathodes cycled at 4.4 V. When the operation voltage exceeds 4.5 V, the lattice strain will further exacerbate LRNDs decomposition, accompanied by substantial oxygen release and mechanical degradation in form of intragranular cracks (strain relief), thus accelerating the voltage and capacity decay. The comprehensive findings affirm that the Co-free component design, when combined with a single-crystal morphology, has detrimental effects rather than beneficial ones on the electrochemical performance of layered cathode materials. To improve the cycle stability of low-cost Co-free SC cathodes, it is crucial to eliminate the formation of LRNDs. One promising strategy involves introducing non-magnetic substitutions (such as $Al^{3+}$) into Co-free cathodes to alleviate magnetic frustration and suppress LRNDs formation, potentially resolving their cyclability issue.

## Discussion

In summary, while the removal of Co from SC cathodes offers cost benefits, it introduces a distinctive nanodomain $Li_2TMO_3$ phase that heterogeneously distributes within the bulk structure of Co-free SC layered cathode materials. The emergence of the $Li_2TMO_3$ nanodomain alleviates magnetic frustration within the TM layer but results in unsatisfactory electrochemical performance. Significant strain evolution and phase degradation in bulk occur during cycling, resulting from the distinct electrochemical behaviors between $Li_2TMO_3$ and $LiTMO_2$ phases. These compelling findings affirm that Co removal from SC layered materials induces subtle structural changes that compromise the material's performance in LIBs. In contrast to polycrystalline cathodes, where Co-free designs have seen some successes, direct Co-removal strategies in SC cathode systems are deemed undesirable. Therefore, new strategies must be devised to strike a balance between cost, sustainability, and performance in SC cathodes to realize their promising role in advancing vehicle electrification.

## Methods

### Materials synthesis

Co-precipitation methods were used to synthesize precursors for Co-free SC cathodes. $Ni_{0.75}Mn_{0.25}(OH)_2$, $Ni_{0.80}Mn_{0.20}(OH)_2$ and $Ni_{0.90}Mn_{0.10}(OH)_2$ precursors were produced by first adding $NiSO_4·6H_2O$ (Sigma-Aldrich, ≥98%) and $MnSO_4·H_2O$ (Sigma-Aldrich, ≥99%) with appropriate ratios into DI water and mixing to obtain a uniform 2 M metal ion solutions. The solution was then pumped into a

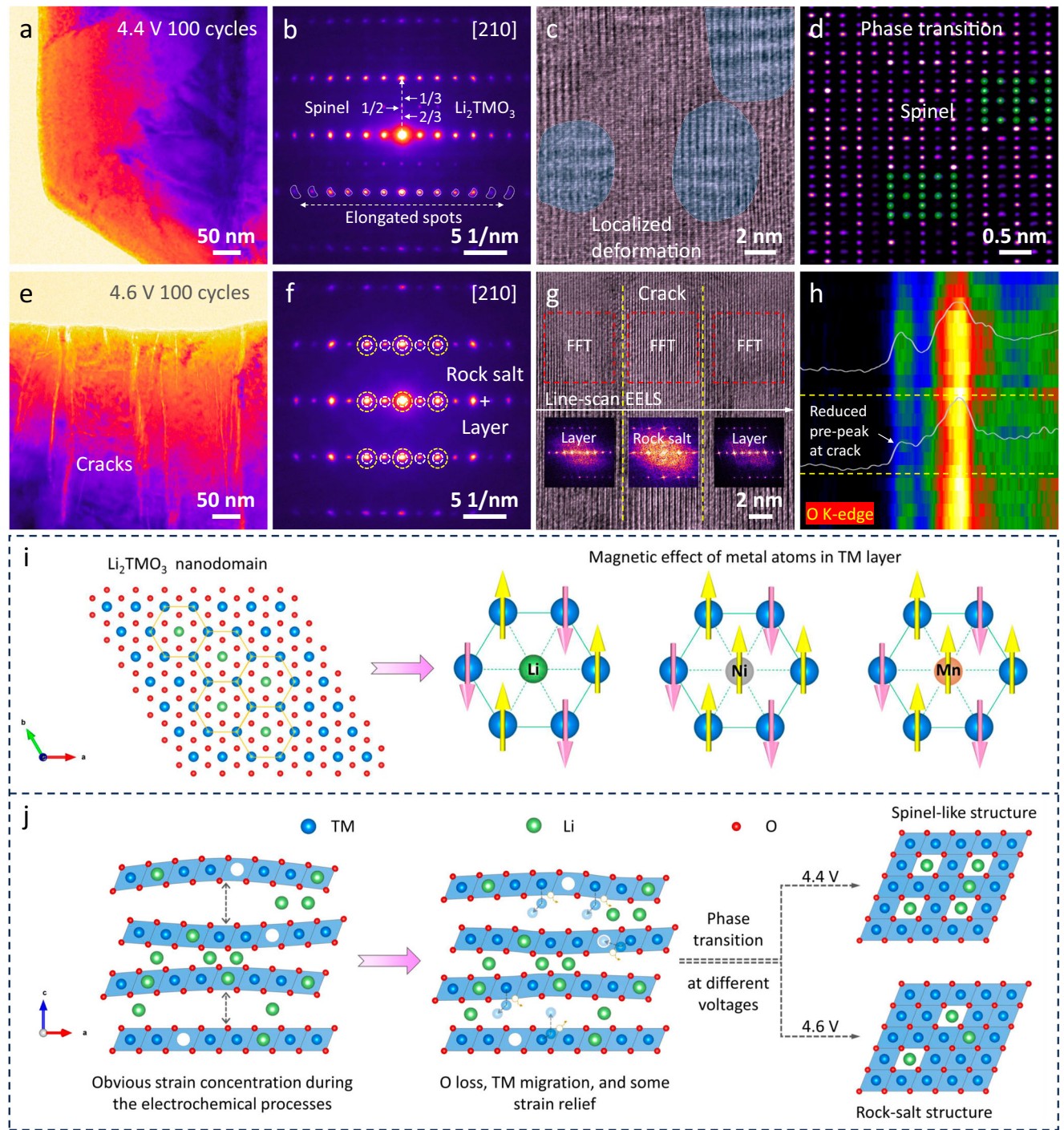

**Fig. 6 | Microscopic TEM observation for cycled SC75 and the formation and degradation mechanism of LRNDs. a** Low-magnification TEM image of SC75 particle after 100 cycles at 0.5 C (100 mA g⁻¹) and 2.8–4.4 V. **b** Corresponding SAED pattern showing the weak Li₂TMO₃ phase and newly formed spinel diffraction spots. **c** HRTEM image of SC75 in bulk area. **d** Enlarged HRTEM image showing spinel phase transition. **e** Low-magnification TEM image of SC75 particle after 100 cycles at 0.5 C (100 mA g⁻¹) and 2.8–4.6 V where severe particle cracking and degradation is visible. **f** Corresponding SAED pattern where newly formed rock salt structures are present. **g** HRTEM image showing the structural transition on the crack. **h** EELS line scan of O K-edge along the marked direction in (**g**). **i** The formation mechanism of LRNDs in Co-free SC layered cathode. **j** Phase transition routes during electrochemical process.

4 L batch reactor under an $N_2$ atmosphere. Meanwhile, 4.0 mol L⁻¹ NaOH solution (aq.) and 5.0 mol L⁻¹ NH₄OH solution (aq.), which acted as precipitating and chelating agents, were also pumped into the reactor. During the co-precipitation process, the pH value was controlled to 11.5, the temperature kept at 60 °C, and the stirring speed maintained at 1000 rpm. After the reaction, the precursor powders were obtained by filtering, washing, and vacuum drying in an oven overnight. Then, the precursors were thoroughly mixed with

LiOH·H₂O (Li:TM = 1.03:1) and calcined under O₂ flow. SC75 was obtained by sintering Ni₀.₇₅Mn₀.₂₅(OH)₂ at 960 °C for 2 h and 900 °C for 10 h at the heating rate of 5 °C min⁻¹. SC80 was obtained by sintering Ni₀.₈₀Mn₀.₂₀(OH)₂ at 940 °C for 2 h and 870 °C for 10 h at the same heating rate. SC90 was obtained by sintering Ni₀.₉₀Mn₀.₁₀(OH)₂ at 920 °C for 2 h and 870 °C for 10 h at the same heating rate. Finally, the powders were carefully grounded to obtain SC products.

## Electrochemistry tests

For electrode preparation, active materials were mixed with carbon black (C45 Conductive Carbon Black, TIMCAL) and polyvinylidene fluoride (PVDF, Solvay® 5130 PVDF binder dissolved in n-methyl-2-pyrrolidone (NMP)) at 80:10:10 wt% ratios. The mixture was then ground in a mortar at 2000 rpm for 9 min (3 min per cycle), and the obtained slurry was cast on the Al foil as the cathode. After drying at 80 °C under vacuum for 12 h to remove traces of solvent, 2032-type coin cells were assembled using the prepared cathode with a diameter of 14 mm (mass loading of active material is ~5.2 mg cm$^{-2}$), a Li metal foil (99.9% purity, MTI. 16.0 mm × 0.6 mm, diameter × thickness) as anode, a Celgard 2325 separator (25 μm), and 1.2 M LiPF$_6$ in EC/EMC (3:7) electrolyte (GEN II with <20 ppm water content, 40 μl). The cells were then cycled between 2.8 and 4.4 or 4.6 V vs Li$^+$/Li in a temperature-controlled chamber at 25 °C ± 2 °C. The C-rate is defined as 1 C = 200 mA g$^{-1}$.

## Synchrotron X-ray diffraction

HEXRD was conducted at 17-BM beamline of the Advanced Photon Source (APS) at Argonne National Laboratory, with an average wavelength of 0.24085Å. Rietveld refinements were carried out using GSAS software packages.

In situ HEXRD measurements were performed at 11-ID-C beamline ($\lambda = 0.1173$Å) of the APS. The high penetration and low absorption of HEXRD are beneficial for observing tiny phase changes that are usually invisible from lab-scale XRD. The 2032-coin cells with two 3-mm holes were assembled for X-rays to pass through, and the holes were sealed with Kapton tape to prevent exposure to air. The mass loading of active material in electrode is ~5.2 mg cm$^{-2}$. The galvanostatic charged-discharged tests were carried out on MACCOR battery system. Diffraction patterns were collected once every 10 min.

## X-ray absorption spectroscopy

XAS experiments were carried out at 7-BM-B beamline of National Synchrotron Light Source II (NSLS-II) at Brookhaven National Laboratory. The XAS data obtained was normalized and analyzed using the Athena and Artemis software packages

## 3D-continuous rotation electron diffraction

3D-CRED tests were performed on the FEI Talos F200X TEM. First, the SC particles were dispersed with ethanol and then dropped onto the lacey carbon grid. A Fischione Model 2550 Cryo Transfer Tomography Holder was used to load the grid into the TEM. In the 3D-CRED experiment, the sample was adjusted to the eccentric height and started with a single tilt from −45° to 45° under the control of a Python script. The rotation rate was 0.6° s$^{-1}$. At the same time, a video of the diffraction pattern (100 ms per frame) was captured using the Velox software. The acquired data was then converted into MRC format and processed using the *REDprocessing software package*[50].

## Scanning electron microscopy and transmission electron microscopy

SEM measurement was conducted on JEOL JSM-7100F. TEM, SAED, EDS, and HRTEM were conducted using the Argonne chromatic aberration-corrected TEM (ACAT) (a FEI Titan 80–300ST with an image aberration corrector to compensate for both spherical and chromatic aberrations) at an accelerating voltage of 200 kV. For TEM tests, thin-section TEM specimens were prepared using a commercial Gatan precision ion polishing system. For the cycled samples, the coin cells were first disassembled in a glove box with an Ar atmosphere. The obtained cathode electrodes were washed immediately using dimethyl carbonate and then completely dried under a vacuum. The dried samples were also thinned like the pristine sample.

## Scanning X-ray diffraction microscopy

SXDM experiments were performed on the CNM-APS 26-ID-C hard x-ray nanoprobe beamline of APS. The 10 keV beam was focused using a 160 um Fresnel Zone Plate with an outmost zone width of 30 nm. Diffraction images were collected on an Eiger 2×1 M detector at a distance of 1 m from the sample. The measurement was performed under a high vacuum to prevent further oxidation of the sample. For strain analysis, selected areas were raster scanned using a focused beam with a spatial resolution of 50 nm. The raster scan was repeated at 16 different rocking curve angles spanned over 3°. With each scan, a two-dimensional diffraction pattern is collected at the scanning point on the area detector. In this way, it is equivalent to obtaining a three-dimensional reciprocal space map about the (003) reflection for each point scanned on the particle. Data analysis was performed using customized Python scripts. Dark field images shown in the work are the processed results from the reduction of five-dimensional datasets (three dimensions in reciprocal space and two dimensions in real space).

## Full-field transmission X-ray microscopy imaging

TXM imaging was conducted at the 18-ID FXI beamline of NSLS-II at Brookhaven National Laboratory to acquire 3D nano-XANES datasets. At each energy, 3D tomography was reconstructed from the projection images taken from 0 to 180°. The voxel resolution of X-ray microscopy imaging was 40 nm.

## Data availability

The data that support the findings of this study are available from the corresponding authors upon request.

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

## Acknowledgements

Work performed at the Center for Nanoscale Materials, a U.S. Department of Energy Office of Science User Facility, was supported by the U.S. DOE, Office of Basic Energy Sciences, under Contract No. DE-AC02-06CH11357. This work gratefully acknowledges support from the U.S. Department of Energy (DOE), Office of Energy Efficiency and Renewable Energy, Vehicle Technologies Office. Argonne National Laboratory is operated for DOE Office of Science by UChicago Argonne, LLC, under contract number DE-AC02-06CH11357. This research used resources of the Advanced Photon Source (17-BM, 11-ID-C and 26-ID-C), a U.S. Department of Energy (DOE) Office of Science User Facility operated for the DOE Office of Science by Argonne National Laboratory under Contract No. DE-AC02-06CH11357. Use of the National Synchrotron Light Source II (beamline 7-BM and 18 ID) is supported by the US Department of Energy, an Office of Science user Facility operated by Brookhaven National Laboratory under contract number DE-SC0012704.

## Author contributions

L.Y., T.Liu, and K.A. conceived of and designed the experiments. A.D., X.H., X.O., and T.Liu synthesized all the materials and conducted electrochemical measurements. L.Y. and J.Wen carried out the TEM, 3D-CRED and EELS tests. T.Z. conducted SXDM experiment and analysis. J.Wang, T.Li, L.M., R.A., and S.N.E., performed ex-situ/in-situ synchrotron HEXRD and XAS. W.H., X.X., and M.G. performed TXM and data analysis. L.Y., A.D., J.Wen, T.Liu, and K.A. wrote the manuscript and all authors edited the manuscript.

## Competing interests

The authors declare no competing interests.
