## [Transparent Peer Review file · Nature Communications]

Parasitic Structure Defect Blights Cobalt-Free Single Crystalline Cathodes

Corresponding Author: Dr Khalil Amine

Version 0:

Reviewer comments:

Reviewer #1

(Remarks to the Author)

In this work, the authors studied the effect of Co removal on the electrochemical cycling stability of the single crystalline layered cathode material. It was found the cycling stability of the Co-free single crystalline material is worse than expected. A voltage decay issue was observed similar to the Li-excess cathode material. Multi scales characterizations were applied to explore the structural transformation after long-term cycling. A unique Li_2MnO_3 nanodomain was confirmed to induce cycling instability and voltage decay. These new insights can serve as design principles for future discovery of improved energy density for LIBs.

In regard to this manuscript, the reviewer raises the following questions:

1. The authors need to clearly explain the role of Co removal for Li_2MnO_3 nanodomain formation.
2. In order to clearly demonstrate the Co-contained single crystal sample does not have the Li_2MnO_3 nanodomain, another set of controlled sample and characterizations are necessary.
3. It was claimed the magnetic frustration plays a significant role in the Li-excess domain, but there are no calculation results to support this hypothesis.
4. Since TEM technique can only provide local information, the authors shall show multiple particles results for each of the case in the supplementary information.
5. Can the authors provide some potential strategies to mitigate the voltage decay issues? If the mechanism understanding is correct, it should provide hints for engineering modifications. It will be ideal that the author can even demonstrate the performance improvement based on one of the proposed strategies.

Reviewer #2

(Remarks to the Author)

Cobalt-free layered cathode has emerged as a research focus for the sustainable development of lithium-ion batteries, driven by the challenges related to cobalt's limited supply, high costs, and ethical concerns. Nevertheless, the effect of cobalt-free design on single crystal cathodes remains underexplored. In this work, the authors did a rigorous mechanism study on the lattice structure evolution of Co-free single crystal cathodes by using many fancy diagnostic tools. This manuscript insightfully reveals the presence of heterogeneous Li-rich nanodomains within Co-free single crystal has great impacts on the structure and performance degradation, which differs significantly from conventional NMC and polycrystalline cobalt-free layered cathodes. Generally, I welcome these important findings and the combination analysis of multiscale and multidimensional characterization approaches in this work is greatly impressive. Therefore, I strongly encourage its publication in Nature Communications, though there are some minor points where clarification or completion is recommended:

1. Please provide evidence of compositional analysis to prove that the sample was well synthesized.
2. It is interesting that the authors used a 3D-CRED technique to fast detect the overall structural change of the single crystal particles. Can the authors provide a detailed description of how to conclude stacking faults based on streak diffraction?
3. This manuscript only shows 3D-CRED results of one particle. It is recommended that the author provide additional data to support this finding.
4. What does the channeling effect in the HRTEM image of Fig. 3i refer to? How is it formed in TEM imaging? Please give

more explanations.

5. In Fig. 6b, why do elongated higher-order diffraction spots reflect strain inside the particle?

Reviewer #3

(Remarks to the Author)

Reviewer #1: The author unveiled that removing Co from SC cathodes is structurally and electrochemically unfavorable, leading to an unusual voltage fade issue that was rarely observed in other Ni-rich cathodes. By combining multiscale diffraction and imaging techniques, it is found that, lithium-rich nanodomain as a heterogeneous phase, universally exists in the lattice, resulting in fast voltage and capacity degradation. The data has a high level of visual appeal. Thus, I recommend this work to publish in Nature Communication with some major revisions.

1. The aim of this article was to evaluate the effectiveness of cobalt in single-crystalline cathodes. However, the main focus of the article is on cobalt-free samples ($\text{LiNi}_{0.75}\text{Mn}_{0.25}\text{O}_2$). Therefore, synthesizing a cobalt-containing sample that doesn't exhibit rapid voltage fade is essential to support your argument. However, based on my knowledge, cobalt-containing single-crystalline samples such as $\text{LiNi}_{0.76}\text{Mn}_{0.14}\text{Co}_{0.1}\text{O}_2$ still demonstrate noticeable capacity and voltage fade.^{1,2} Additionally, it is crucial to demonstrate the absence of LRNDs in the cobalt-containing sample.

2. The reactivity of oxygen in conventional layered oxides like LiNiO_2 occurs due to depopulation of the highly hybridized O 2p states alongside metal 3d states. This reactivity leads to the generation of O_2 gas when the cathode is charged to ultrahigh voltage, resulting in the formation of spinel or disordered rocksalt phases. It is erroneous to attribute the rapid voltage decay in the SC75 sample cycled at high cut-off voltage solely to the absence of cobalt. This is supported by the observed deterioration in voltage fade in any cobalt-containing layered oxides, such as LiCoO_2 or $\text{LiNi}_{0.95}\text{Co}_{0.05}\text{O}_2$, when charged to high voltage.

3 Minor: extra white line in Fig.2a.

4. The chemical stoichiometry of Li:TM in the precursor is 1.03:1, so it is reasonable to find Li_2MnO_3 nanodomains in this layered material ($x\text{LiNiO}_2 \cdot (1-x)\text{Li}_2\text{MnO}_3$). How to evaluate the importance of the existence of nanodomains in SC? The reversible capacity of SC75 only increased by 10mAh/g when the cutoff potential was raised from 4.4V to 4.6V. I'm just wondering if this phenomenon will also occur in Co-containing samples? In other words, LRNDs show almost no effect when the sample is cycled at 2.8V-4.4V? So, this phenomenon is not very crucial for the advancement in SC cathodes.

1. Bi, Y., Tao, J., Wu, Y., Li, L., Xu, Y., Hu, E., Wu, B., Hu, J., Wang, C., Zhang, J.-G., et al. (2020). Reversible planar gliding and microcracking in a single-crystalline Ni-rich cathode. *Science* 370, 1313-1317. doi:10.1126/science.abc3167.

2. Ryu, H.-H., Namkoong, B., Kim, J.-H., Belharouak, I., Yoon, C.S., and Sun, Y.-K. (2021). Capacity Fading Mechanisms in Ni-Rich Single-Crystal NCM Cathodes. *Acs Energy Lett* 6, 2726-2734. 10.1021/acseenergylett.1c01089.

Version 1:

Reviewer comments:

Reviewer #1

(Remarks to the Author)

In the revised manuscript, the authors have clearly explained the magnetic frustration effect and its relationship with the Li_2MnO_3 domain formation. More importantly, a strategy following this hypothesis has been proposed for improving the cycling stability of the Co-free single crystalline materials. Most of the questions from the previous reviewers have been carefully addressed. Just a minor suggestion to include at least the Al doping or zero-spin element doping as a perspective in the current work as the authors would like to publish the actual results in a separate work. The electrochemical performances shown in Figure R7 are based on full cells or half cells? What is the C-rate for the testing and what is the capacity retention after 500 cycles?

Reviewer #2

(Remarks to the Author)

All of the concerns have been addressed in detail in the revisions, and the quality of the manuscript has been further improved. The reviewer recommends this manuscript for publication in Nature Communications now.

Reviewer #3

(Remarks to the Author)

Authors revised the manuscript according the comments. I have no any new commrnt at this stage and recommend the acceptance of the manuscript in the Nature Communications without further revisions.

Itemized Responses/Revisions to the original manuscript

Article Reference: NCOMMS-24-14403-T

Manuscript Title: “Parasitic Structure Defect Blights Sustainability of Cobalt-Free Single Crystalline Cathodes”

Responses to reviewer 1:

General comment: “In this work, the authors studied the effect of Co removal on the electrochemical cycling stability of the single crystalline layered cathode material. It was found the cycling stability of the Co-free single crystalline material is worse than expected. A voltage decay issue was observed similar to the Li-excess cathode material. Multi scales characterizations were applied to explore the structural transformation after long-term cycling. A unique Li_2MnO_3 nanodomain was confirmed to induce cycling instability and voltage decay. These new insights can serve as design principles for future discovery of improved energy density for LIBs.”

Response: We would like to thank the reviewer for his/her positive feedback and valuable suggestions, which definitely help us make this work more solid and appealing. Based on the reviewer’s suggestions, we have carefully revised this manuscript. Below is point-by-point response to the comments.

1. Comment #1

Comment: “The authors need to clearly explain the role of Co removal for Li_2MnO_3 nanodomain formation.”

Response: Thanks for your suggestion. In the revision, we highlight the role of Co removal for Li_2MnO_3 nanodomain formation. The revised description is as follow:

The formation of the Li_2MnO_3 , where Li is present in the transition metal (TM) layer, can attributed to the magnetic interactions within TM layers and the rigorous calcination conditions necessary for single crystalline cathode synthesis. The magnetic interaction mechanism has recently been recognized as critical driving forces in tuning the atomic occupancies of TM and Li ions in layered cathode materials. In a typical NMC cathode, the spins of TM ions construct a two-dimensional triangular network. Within this network, magnetic Ni^{2+} and Mn^{4+} ions will induce strongly frustrated magnetic interactions in the TM layers, giving rise to lattice instability. According to the previous report (Figure R1)^[1], nonmagnetic

Co^{3+} ions can form a nonmagnetic center on a honeycomb lattice unit, stabilizing the surrounding TM spins in an antiferromagnetic arrangement. Similarly, Li ions which also has no spin, can occupy similar positions to Co^{3+} in TM layers to alleviate magnetic frustration and form a stable cathode structure. As a result, in the case of Co-free single crystalline layered cathode studied here, some Li ions preferentially occupies positions in the TM layers to release the strong magnetic frustration due to Co's absence. Therefore, removing Co generally facilitates the formation of Li_2MnO_3 nanodomains. On the other hand, the increased calcination temperature and grain size greatly exacerbate the uneven distribution of Li in the single crystalline particles. The combination of these two aspects finally leads to the formation of the LRNDs.

This revised description has been updated in the revised manuscript.

[1] M. Li, J. Lu, Science 2020, 367, 979.

Figure R1. Magnetic frustration of Ni, Co, and Li in the TM layer^[1].

2. Comment #2

Comment: “In order to clearly demonstrate the Co-contained single crystal sample does not have the Li_2MnO_3 nanodomain, another set of controlled sample and characterizations are necessary.”

Response: This is a good suggestion. In our previous report, the 3D-CRED results of Co-containing single crystalline $\text{LiNi}_{0.83}\text{Mn}_{0.06}\text{Co}_{0.11}\text{O}_2$ have shown the absence of Li_2MnO_3 nanodomains (Figure R2)^[2].

To confirm this, we further conducted 3D-CRED and 2D-SAED experiments on Co-containing single crystalline $\text{LiNi}_{0.81}\text{Mn}_{0.06}\text{Co}_{0.13}\text{O}_2$ sample with slightly changed compositions. As shown in Figure R3, the 3D-CRED results display typical layered LiTMO_2 diffraction lattices without extra structural defects detected. In addition, as shown in Figure R4, the 2D-SAED results show no obvious streak diffraction features. These collective evidences indicate that the Co-containing single crystalline samples are devoid of Li_2MnO_3 nanodomains. We have added these results in the revised supporting information (Supplementary Fig. 7 and Supplementary Fig. 11).

[2] W. Huang, T. Liu, L. Yu, J. Wang, T. Zhou, J. Liu, T. Li, R. Amine, X. Xiao, M. Ge, L. Ma, S. N. Ehrlich, M. V. Holt, J. Wen, K. Amine, *Science* 2024, 384, 912.

Figure R2. The 3D-CRED results of Co-containing single crystalline $\text{LiNi}_{0.83}\text{Mn}_{0.06}\text{Co}_{0.11}\text{O}_2$ ^[2].

Figure R3. The 3D-CRED results of Co-containing single crystalline $\text{LiNi}_{0.81}\text{Mn}_{0.06}\text{Co}_{0.13}\text{O}_2$, showing typical layered LiTMO_2 diffraction lattice without extra structural defects.

Figure R4 The SAED results of Co-containing single crystalline $\text{LiNi}_{0.81}\text{Mn}_{0.06}\text{Co}_{0.13}\text{O}_2$ along the [210] zone axis, showing no obvious streak diffraction.

3. Comment #3

Comment: *“It was claimed the magnetic frustration plays a significant role in the Li-excess domain, but there are no calculation results to support this hypothesis.”*

Response: Thanks for your comments. Magnetic frustration has been demonstrated as an critical driving force causing the lattice instability of layered LiTMO₂ cathode materials, which has been verified by numerous theoretical calculations and experiments^[3-7]. Introducing nonmagnetic ions like Li in the Co-free cathodes is effective to release magnetic frustration. In previous reports, our colleagues demonstrated the low formation energy of honeycomb structure with Li center in the TM layer of Ni-rich Co-less cathode^[4]. Furthermore, it has been predicted by first-principles calculations that the TM layer ordering of LiNi_{0.5}Mn_{0.5}O₂ transforms from the zigzag structure to the honeycomb structure (Li occupation) during heating^[6]. For the Co-free single crystalline samples, the higher-temperature calcination (>900 °C) further facilitates the occupation of Li in the TM layer, finally leading to the formation of the Li-excess domain. In this work, we first reveal the presence of these microscopic defects and confirm their crucial influence on electrochemical performances using advanced characterization techniques. Extensive characterization results about Co-free single crystalline samples of different fractions (SC75, SC80 and SC90) and the absence of LRNDs in Co-containing samples fully support our proposed magnetic frustration mechanism. The above-mentioned reports have been cited in the revised manuscript.

[3] J. Zheng, Y. Ye, T. Liu, Y. Xiao, C. Wang, F. Wang, F. Pan, *Acc. Chem. Res.* 2019, 52, 2201.

[4] J. Zheng, G. Teng, C. Xin, Z. Zhuo, J. Liu, Q. Li, Z. Hu, M. Xu, S. Yan, W. Yang, F. Pan, *J. Phys. Chem. Lett.* 2017, 8, 5537.

[5] Y. Xiao, T. Liu, J. Liu, L. He, J. Chen, J. Zhang, P. Luo, H. Lu, R. Wang, W. Zhu, Z. Hu, G. Teng, C. Xin, J. Zheng, T. Liang, F. Wang, Y. Chen, Q. Huang, F. Pan, H. Chen, *Nano Energy* 2018, 49, 77.

[6] Y. Hinuma, Y. S. Meng, K. Kang, G. Ceder, *Chem. Mater.* 2007, 19, 1790.

[7] N. A. Chernova, M. Ma, J. Xiao, M. S. Whittingham, J. Breger, C. P. Grey, *Chem. Mater.* 2007, 19, 4682.

4. Comment #4

Comment: *“Since TEM technique can only provide local information, the authors shall show multiple particles results for each of the case in the supplementary information.”*

Response: Thanks for your advice. We have further supplied 3D-CRED results of multiple randomly selected SC75 particles in the revised supplementary information (Supplementary Fig. 6). As shown in Figure R5, all these 3D-CRED projection patterns present the extra triple period along the direction of $(\bar{1}20)^*$, suggesting the existence of stacking defects. Furthermore, the HRTEM characterization of another

SC75 particle (Supplementary Fig. 10) as well as the extensive SAED results of SC80 (Supplementary Fig. 12) and SC90 (Supplementary Fig. 13) samples together support our conclusions.

Figure R5. The 3D-CRED results of other randomly selected SC75 particles.

5. Comment #5

Comment: “Can the authors provide some potential strategies to mitigate the voltage decay issues? If the mechanism understanding is correct, it should provide hints for engineering modifications. It will be ideal that the author can even demonstrate the performance improvement based on one of the proposed strategies.”

Response: Thank you for your insightful suggestion. We have indeed explored an effective method to address the voltage fade issue rooted in Li-excess domains. In this work, we identified the Li-excess domain as a primary cause of structural and electrochemical deterioration. Therefore, suppressing the formation of heterogeneous phases is necessary to improve electrochemical performance. According to the magnetic frustration theory, introducing non-magnetic substitute elements is a feasible approach. We introduced Al into Co-free cathodes to alleviate magnetic frustration and mitigate the formation of Li-excess domains. As shown in the TEM and SEAD images (Figure R6), the Al-doped Co-free single-crystalline cathode displays a significantly reduced Li-excess phase. Electrochemical tests also indicate substantial increases in voltage and cycle stability (Figure R7). This result in turn indicates the detrimental effects of Li-excess domains. We are currently preparing a research paper to further elaborate on these findings.

Figure R6. The TEM and EDS characterizations of Al-doped Co-free single-crystalline cathode.

Figure R7. Electrochemical performances of Al-doped Co-free single-crystalline cathode.

Responses to reviewer 2:

General comment: “Cobalt-free layered cathode has emerged as a research focus for the sustainable development of lithium-ion batteries, driven by the challenges related to cobalt’s limited supply, high costs, and ethical concerns. Nevertheless, the effect of cobalt-free design on single crystal cathodes remains underexplored. In this work, the authors did a rigorous mechanism study on the lattice structure evolution of Co-free single crystal cathodes by using many fancy diagnostic tools. This manuscript insightfully reveals the presence of heterogeneous Li-rich nanodomains within Co-free single crystal has great impacts on the structure and performance degradation, which differs significantly from conventional NMC and polycrystalline cobalt-free layered cathodes. Generally, I welcome these important findings and the combination analysis of multiscale and multidimensional characterization approaches in this work is greatly impressive. Therefore, I strongly encourage its publication in Nature Communications, though there are some minor points where clarification or completion is recommended.”

Response: Thank you so much for the positive comments on the significance of this work and the valuable suggestions. We believe these suggestions significantly improve the quality of the article. We responded to the comments point by point as follows.

6. Comment #1

Comment: “Please provide evidence of compositional analysis to prove that the sample was well synthesized.”

Response: Thanks for your suggestion. We have added the STEM-EDS results of SC75 sample in the revised supporting information (Supplementary Fig. 1). As shown in Figure R8, the STEM-EDS results show that the sample was well synthesized with an ideal compositional ratio.

Figure R8. EDS element analysis of the SC75 sample.

7. Comment #2

Comment: *“It is interesting that the authors used a 3D-CRED technique to fast detect the overall structural change of the single crystal particles. Can the authors provide a detailed description of how to conclude stacking faults based on streak diffraction?”*

Response: Thanks for your question. The reciprocal space of a two-dimensional plane is a one-dimensional rod. Stacking faults are two-dimensional planar defects, so the truncation rod of stacking faults results in diffraction streaking. This Ewald sphere intersects with the truncate rod when taking a series of tilt diffraction patterns for 3D-CRED. With the reconstruction of the 3D diffraction pattern, we can also reconstruct the truncation rod. This is why 3D-CRED is an effective method for quickly detecting overall structural changes in primary particles.

8. Comment #3

Comment: *“This manuscript only shows 3D-CRED results of one particle. It is recommended that the author provide additional data to support this finding.”*

Response: Thanks for your kind advice. We have further supplied 3D-CRED results of multiple randomly selected SC75 particles in the revised supplementary information (Supplementary Fig. 6). As shown in Figure R9, all these 3D-CRED projection patterns present the extra triple period along the direction of $(\bar{1}20)^*$, suggesting the existence of stacking defects and ruling out the occasionality.

Figure R9. The 3D-CRED results of other randomly selected SC75 particles.

9. Comment #4

Comment: *“What does the channeling effect in the HRTEM image of Fig. 3i refer to? How is it formed in TEM imaging? Please give more explanations.”*

Response: This image was taken using a spherical and chromatic aberrations corrected microscope. When both spherical and chromatic aberrations are corrected close to zero, an amplitude contrast imaging (ACI) condition is achieved (ref: Wen, J. G., et al. "Amplitude contrast imaging: High resolution electron microscopy using a spherical and chromatic aberration corrected TEM." *Microscopy and Microanalysis* 20.S3 (2014): 942-943.). Under ACI imaging conditions, the intensity at each atomic column is dominated by the amplitude instead of the phase in the conventional spherical aberration-corrected HRTEM images. Under ACI conditions, atomic resolution channeling contrast (ARCC) also can be realized as described in the paper (ref: A. Wang, F. R. Chen, S. Van Aert, D. Van Dyck, *Ultramicroscopy* 110, 527–534 (2010)). In channeling theory, high energy electron travels along atomic columns. Each atom in the column behaves as a mini-lens to bend high-energy electrons, such that light elements bend electron less compared to heavy elements. Therefore, the intensity variation in the same type of atomic columns reflects the occupation rate change.

10. Comment #4

Comment: *“In Fig. 6b, why do elongated higher-order diffraction spots reflect strain inside the particle?”*

Response: Thanks for your good question. Elongated diffraction spots in higher-order diffraction spots indicate changes in diffraction length and angle, resulting from the lattice strain in the particle.

Responses to reviewer 3:

General comment: “The author unveiled that removing Co from SC cathodes is structurally and electrochemically unfavorable, leading to an unusual voltage fade issue that was rarely observed in other Ni-rich cathodes. By combining multiscale diffraction and imaging techniques, it is found that, lithium-rich nanodomain as a heterogeneous phase, universally exists in the lattice, resulting in fast voltage and capacity degradation. The data has a high level of visual appeal. Thus, I recommend this work to publish in Nature Communication with some major revisions.”

Response: The authors would like to thank the reviewer for his/her time and effort. In addition, we appreciate his/her positive feedback and great suggestions. The suggested comments were answered point by point in the following section.

11. Comment #1

Comment: “The aim of this article was to evaluate the effectiveness of cobalt in single-crystalline cathodes. However, the main focus of the article is on cobalt-free samples ($\text{LiNi}_{0.75}\text{Mn}_{0.25}\text{O}_2$). Therefore, synthesizing a cobalt-containing sample that doesn't exhibit rapid voltage fade is essential to support your argument. However, based on my knowledge, cobalt-containing single-crystalline samples such as $\text{LiNi}_{0.76}\text{Mn}_{0.14}\text{Co}_{0.1}\text{O}_2$ still demonstrate noticeable capacity and voltage fade.^{1,2} Additionally, it is crucial to demonstrate the absence of LRNDs in the cobalt-containing sample.”

Response: Thanks for your comments and suggestions. It is well documented that voltage fade typically results from increased impedance and structural degradation. The reviewer mentioned that the $\text{LiNi}_{0.76}\text{Mn}_{0.14}\text{Co}_{0.1}\text{O}_2$ cathode in the previous report still demonstrates noticeable capacity and voltage fade. According to the explanation in this reference, it attributed the voltage fade to an increase in impedance caused by the interfacial electrolyte decomposition. The interfacial side reactions and surface structural reconstruction have historically been considered responsible for the performance degradation of cathode material. In this study, we emphasize the impact of bulk atomic structure defects on the structural stability of single crystalline cathodes, an area that has been less understood in previous research. Our work shows that cobalt-free single crystalline samples present more prominent voltage fade behavior and cyclic instability. The LRNDs have been identified as underlying factors in inducing bulk structural degradation of Co-free single crystalline samples, which is different from the traditional surface degradation mechanism. As shown in Figure R10, the heterogeneous structures (composed of LiTMO_2 and Li_2TMO_3) with differential electrochemical activities lead to extensive nanoscale strain during cycling. This is also evidenced by the SXDM experiment (Fig. 4, revised manuscript). Furthermore, the HRTEM characterization demonstrated that the LRNDs substantially alter the structural stability and aggravate oxygen release. As shown in Figure R11, the bulk phase transition of spinel occurs at 4.4V-cycling and rock salt occurs at 4.6 V-cycling. The lattice

microstrain and phase transition in the bulk increase the transport barriers of Li^+ ions during the electrochemical process.

To further prove this, we designed Al-doped Co-free cathodes to reduce the LRNDs content, as Al, which is a non-magnetic ion could effectively reduce Li occupancy in TM layers driven by TM layer magnetic frustration (Figure R12). The electrochemical tests indicate significant increases in voltage and cyclic stability (Figure R13). Therefore, it is reasonable to attribute LRNDs to break voltage stability. For clarity, we further optimized the relevant descriptions in the revised manuscript.

Based on the reviewer's suggestion, we further conducted 3D-CRED and 2D-SAED tests on the Co-containing single crystalline $\text{LiNi}_{0.81}\text{Mn}_{0.06}\text{Co}_{0.13}\text{O}_2$ sample. As shown in Figure R14, the 3D-CRED results display typical layered LiTMO_2 diffraction lattices without extra structural defects detected. Furthermore, as shown in Figure R15, the 2D-SAED results show no obvious streak diffraction features. These collective evidences indicate that the Co-containing single crystalline samples are devoid of Li_2MnO_3 nanodomains. These new characterization results have been added in the revised supporting information (Supplementary Fig. 7 and Supplementary Fig. 11).

Figure R10. The high-magnification TEM images of the SC75 cathode after 100 cycles at 0.5C and 2.8-4.4 V.

Figure R11. Microscopic TEM characterization for the cycled SC75 samples.

Figure R12. The TEM and EDS characterizations of Al-doped Co-free single-crystalline cathode.

Figure R13. Electrochemical performances of Al-doped Co-free single-crystalline cathode.

Figure R14. The 3D-CRED results of Co-containing single crystalline $\text{LiNi}_{0.81}\text{Mn}_{0.06}\text{Co}_{0.13}\text{O}_2$, showing typical layered LiTMO_2 diffraction lattice without extra structural defects.

Figure R15 The SAED results of Co-containing single crystalline $\text{LiNi}_{0.81}\text{Mn}_{0.06}\text{Co}_{0.13}\text{O}_2$ along the [210] zone axis, showing no obvious streak diffraction.

12. Comment #2

Comment: “The reactivity of oxygen in conventional layered oxides like LiNiO_2 occurs due to depopulation of the highly hybridized O 2p states alongside metal 3d states. This reactivity leads to the generation of O_2 gas when the cathode is charged to ultrahigh voltage, resulting in the formation of spinel or disordered rocksalt phases. It is erroneous to attribute the rapid voltage decay in the SC75 sample cycled at high cut-off voltage solely to the absence of cobalt. This is supported by the observed deterioration in voltage fade in any cobalt-containing layered oxides, such as LiCoO_2 or $\text{LiNi}_{0.95}\text{Co}_{0.05}\text{O}_2$, when charged to high voltage.”

Response: Thanks for your wonderful comments. We agree with the reviewer that the orbital hybridization between O 2p states and transition metal 3d states in layered LiTMO₂ cathodes greatly affects the stability of the lattice oxygen. Particularly for Co-containing layered oxides, the overlap of the O and Co redox potentials deleteriously promotes a high lattice O activity at high charge potentials, which results in oxygen release and irreversible phase transition. Our previous report demonstrated that polycrystalline Co-free cathodes have better oxygen and structural stability^[8]. It is with this in mind that we further developed the Co-free single crystalline cathodes with high expectations of structural and electrochemical stability. However, the higher calcination temperature and increased grain size exacerbated the uneven distribution of Li during single-crystalline synthesis. It was found that an atomic structure defect, Li-rich nanodomain as a heterogeneous phase, universally exists in the lattice of Co-free single crystalline cathodes. As a result, Co-free single crystalline layered oxides exhibit an extra oxygen loss mechanism during the electrochemical process. Apart from the conventional oxygen loss in LiTMO₂, the anion charge compensation mechanism of Li-rich nanodomains further aggressive the localized oxygen loss at high charge-discharge voltage. As shown in the HRTEM-EELS characterizations of Figure R16, it observes the reduced pre-peak of O K-edge on the bulk phase transition area. We highlight that the presence of LRNDs in cobalt-free single-crystalline cathodes could be more detrimental to voltage stability, as it would trigger more severe lattice oxygen loss during cycling. This is why the Co-free single crystalline layered cathodes exhibit prominent voltage decay. We have modified some descriptions in the revised manuscript for understanding.

Figure R16 The HRTEM image and corresponding line-scan EELS result of 4.6 V-cycled SC75 particle in the bulk.

[8] T. Liu, L. Yu, J. Liu, J. Lu, X. Bi, A. Dai, M. Li, M. Li, Z. Hu, L. Ma, D. Luo, J. Zheng, T. Wu, Y. Ren, J. Wen, F. Pan, K. Amine, *Nat. Energy* 2021, 6, 277

13. Comment #3

Comment: “Minor: extra white line in Fig.2a”

Response: Thanks for your kind reminder. The extra white lines have been removed in the revised manuscript.

14. Comment #4

Comment: *“The chemical stoichiometry of Li:TM in the precursor is 1.03:1, so it is reasonable to find Li_2MnO_3 nanodomains in this layered material ($x\text{LiNiO}_2 \cdot (1-x)\text{Li}_2\text{MnO}_3$). How to evaluate the importance of the existence of nanodomains in SC? The reversible capacity of SC75 only increased by 10mAh/g when the cutoff potential was raised from 4.4V to 4.6V. I'm just wondering if this phenomenon will also occur in Co-containing samples? In other words, LRNDs show almost no effect when the sample is cycled at 2.8V-4.4V? So, this phenomenon is not very crucial for the advancement in SC cathodes.”*

Response: Thanks for your comments. Layered cathode materials are typically sintered with excess Li source (Li:TM ratio>1) to compensate the Li loss caused by Li volatilization, especially at high temperature, but the Li_2MnO_3 nanodomains were not found before. So we believe the excess Li is not the main formation cause of the LRNDs. In response to comment 1, we have included extensive microscopic structural characterizations that show no apparent Li-rich phase in the Co-containing samples. The formation of LRNDs in Co-free single crystalline cathodes mainly stems from the magnetic interactions within TM layers and the rigorous calcination conditions necessary for single-crystal synthesis. The enhanced magnetic frustration due to Co removal facilitates Li disordering in the layered structure. In order to create a stable crystal structure, nonmagnetic Li^+ ions in place of Co^{3+} are additionally introduced into the hexagonal centers of the TM layer to form stabilization centers during synthesis. On the other hand, the higher calcination temperature and increased grain size exacerbate the uneven distribution of Li in the single-crystalline particles. The combination of these two aspects ultimately leads to the formation of the LRNDs.

For Co-free single crystalline samples, the structural heterogeneity (consisting of LiTMO_2 and Li_2TMO_3) will produce significant lattice strains during cycling. Specifically, during the initial delithiation process, Li^+ extraction predominantly occurs in the LiTMO_2 region due to its distinct redox chemistry compared to the Li_2MnO_3 domains. The lattice expansion of LiTMO_2 is partially constrained by the neighboring inactive Li_2MnO_3 domains, leading to the formation of lattice strain. Therefore, even with the electrochemical operation voltage at 4.4V, the Li-rich phase preferentially degrades to spinel under the induction of lattice strain. This bulk structural degradation leads to an increase in impedance, decreasing the electrochemical performance. As a result, the electrochemical performance of Co-free cathodes is inferior to that of Co-containing cathodes, as noted in other reports. Worse still, when electrochemically operated above 4.5 V, the Li-rich phase induces complete rock salt phase transition, particle damage and prominent oxygen loss, further deteriorating the electrochemical performance. These findings reveal that understanding the inherent structural defects and their adverse impacts is crucial for the development of single crystalline cathodes. Eliminating the adverse effects of

Co removal and searching for effective substituted elements are essential for further developing advanced low-cost and sustainable cathodes.

Itemized Responses/Revisions to the original manuscript

Article Reference: NCOMMS-24-14403-T

Manuscript Title: “Parasitic Structure Defect Blights Sustainability of Cobalt-Free Single Crystalline Cathodes”

Responses to reviewer 1:

General comment: “In the revised manuscript, the authors have clearly explained the magnetic frustration effect and its relationship with the Li_2MnO_3 domain formation. More importantly, a strategy following this hypothesis has been proposed for improving the cycling stability of the Co-free single crystalline materials. Most of the questions from the previous reviewers have been carefully addressed. Just a minor suggestion to include at least the Al doping or zero-spin element doping as a perspective in the current work as the authors would like to publish the actual results in a separate work. The electrochemical performances shown in Figure R7 are based on full cells or half cells? What is the C-rate for the testing and what is the capacity retention after 500 cycles?”

Response: We thank the reviewer for his/her time and effort in our work.

We have added the description of Al doping or zero-spin element doping as a potential modification strategy for Co-free single crystalline materials in the revised manuscript (page 14, paragraph 1, lines 9-13). The detailed description follows: “To improve the cycle stability of low-cost Co-free SC cathodes, it is crucial to eliminate the formation of LRNDs. One promising strategy involves introducing non-magnetic substitutions (such as Al^{3+}) into Co-free cathodes to alleviate magnetic frustration and suppress LRNDs formation, potentially resolving their cyclability issue.”

In Figure R7, the charge-discharge profiles are based on half cells, and the plot of cyclic performance is based on full cells. The C-rate for the electrochemical testing is 0.5C. The capacity retention of full cell after 500 cycles is about 83.3%.

Responses to reviewer 2:

General comment: “All of the concerns have been addressed in detail in the revisions, and the quality of the manuscript has been further improved. The reviewer recommends this manuscript for publication in Nature Communications now.”

Response: Thanks for your approval of our work.

Responses to reviewer 3:

General comment: "Authors revised the manuscript according the comments. I have no any new comment at this stage and recommend the acceptance of the manuscript in the Nature Communications without further revisions."

Response: Thanks for your approval of our work.